# Energy Efficiency Optimization for SWIPT-Enabled IoT Network with Energy Cooperation

**DOI:** 10.3390/s22135035

**Published:** 2022-07-04

**Authors:** Yang Cao, Ye Zhong, Chunling Peng, Xiaofeng Peng, Song Pan

**Affiliations:** 1Periodical Agency, Chongqing University of Technology, Chongqing 400054, China; 2School of Electrical and Electronic Engineering, Chongqing University of Technology, Chongqing 400054, China; zhongye@stu.cqut.edu.cn (Y.Z.); chunlingp@cqut.edu.cn (C.P.); pxf@cqut.edu.cn (X.P.); pansong@2020.cqut.edu.cn (S.P.)

**Keywords:** SWIPT, IoT, energy efficiency, power allocation, time switching, energy cooperation

## Abstract

As an advanced technology, simultaneous wireless information and power transfer (SWIPT), combined with the internet of things (IoT) devices, can effectively extend the online cycle of the terminal. To cope with the fluctuation of energy harvesting by the hybrid access points (H-AP), the energy cooperation base station is introduced to realize the sharing of renewable energy. In this paper, we study the SWIPT-enabled IoT networks with cooperation. Our goal is to maximize the energy efficiency of the system, and at the same time, we need to meet the energy harvesting constraints, user quality of service (QoS) constraints and transmission power constraints. We jointly solve the power allocation, time switching and energy cooperation problems. Because this problem is a nonlinear programming problem, it is difficult to solve directly, so we use the alternating variable method, the iterative algorithm is used to solve the power allocation and time switching problem, and the matching algorithm is used to solve the energy cooperation problem. Simulation results show that the proposed algorithm has obvious advantages in energy efficiency performance compared with the comparison algorithm. At the same time, it is also proved that the introduction of energy cooperation technology can effectively reduce system energy consumption and improve system energy efficiency.

## 1. Introduction

The development of 5G technology has contributed to the rapid spread of the internet of things (IoT). As an important application scenario for 5G, IoT connects a large number of physical objects such as wearable devices, smart home sensors, industrial sensors and agricultural sensors wirelessly to provide various services such as healthcare, smart industry and smart agriculture [1,2]. According to Cisco, the number of global IoT devices in IoT networks will reach 14.7 billion by 2023 [3], which means that the number of wireless devices and traffic demand will grow at an explosive rate, and it is clear that simply using macro base stations in cellular networks to connect this huge number of devices is not enough. To address this problem, dense hybrid access point (H-AP) deployment is seen as a promising technology to meet the quality of service (QoS) needs of devices [4].

The dense deployment of H-APs increases the energy consumption of communication systems, and in the context of a dual carbon strategy, green solutions need to be developed to reduce network-wide energy consumption. To address this challenge, a large number of scholars have explored efficient energy utilization and solutions, mainly in two dimensions: “open source” and “cost reduction”. Specifically, the introduction of renewable energy, i.e., “open source”, such as solar and wind energy, to reduce energy consumption on the grid. Efficient energy management strategies, i.e., “cost reduction”, such as wireless resource management strategies or H-AP sleeping, control the transmitting power of base stations to reduce system energy consumption.

Energy harvesting technology, as an environmentally friendly and economically friendly technology, has been widely used in the communications sector. For example, about 2/3 of the H-APs deployed by China Mobile in Tibet are powered by renewable energy [5]. Huawei has designed solar cellular H-APs around the world to the tune of 20 million kWh [6]. A number of studies have been carried out for communication networks with energy harvesting by combining both “open source” and “cost reduction” dimensions. The authors of [7] propose a dynamic energy-aware power allocation algorithm based on Lyapunov optimization to maximize system throughput. The work in [8] considers the mixed integer programming problem of user association and power allocation to improve the system energy efficiency using Lagrangian algorithms. The authors of [9] propose a joint optimized power allocation and energy management maximization energy efficiency method based on the Lyapunov framework for dense base station networks with energy harvesting, which improves the overall system throughput and optimizes the system energy efficiency. In [10], a generalized Benders decomposition method based on Lagrange multipliers is proposed for dormant networks with energy harvesting and base stations to optimize the transmit power consumption and reduce the energy consumption of the system. However, the renewable energy harvested by base stations in different climates and geographical locations varies significantly due to its stochastic and unstable nature.

To address the unevenness of harvested energy, various approaches have been proposed, mainly including two major approaches, energy storage systems and energy collaboration [11]. The first option requires large capacity battery storage and there are losses in the use of batteries, and the deployment of large quantities can lead to high costs and environmental pollution. On the other hand, energy cooperation is an important technology based on the development of smart grids, and is considered to be an effective solution to improve the energy utilization of the system energy cooperation technology is based on smart grids as a carrier, through its bi-directional power flow devices to effectively solve the problems of H-AP energy shortage and uneven distribution of renewable energy sources [12,13], and has been studied by scholars in conjunction with energy cooperation networks. In [14], a deep reinforcement learning algorithm is proposed to improve the total system throughput by offline training, taking into account the transmit power, energy harvesting and battery capacity constraints in a fused energy cooperation IoT downlink scenario. The authors of [15] investigate the problem of minimizing energy consumption in H-AP networks with caching and propose a low-complexity hierarchical solution algorithm that effectively reduces system power consumption by optimizing terminal bandwidth and energy cooperation mechanisms. The authors of [16] propose a hybrid energy ratio allocation algorithm, which effectively improves the impact of renewable energy on the communication system and reduces the energy cost of the system.

Another important constraint on the development of IoT is the power supply to the terminals. Wired and battery power cannot effectively solve the problem of energy shortage in terminals, so it is particularly important to study the maintenance of the sustainable operation of terminals. Energy harvesting technology is considered an important means of reducing system energy consumption and extending the operation of devices in order to extend the operating time of terminals on the network. Traditional renewable energy sources such as wind and solar are intermittent and unreliable, and a large number of terminals are deployed indoors where solar energy supply is not applicable [17]. In recent years, a large number of scholars have studied the use of simultaneous wireless information and power transfer (SWIPT) technology to solve the energy shortage problem of wireless communication equipment nodes. The characteristics of this technology are to make full use of the characteristics of radio frequency (RF) signals that carry data information and electromagnetic energy at the same time, and to extend the network operation cycle of communication devices by harvesting the surrounding energy for wireless charging while realizing wireless information transmission [18,19]. To further improve the performance of SWIPT-enabled IoT, many studies have been conducted extensively from the scheme of resource allocation. The authors of [20] jointly optimized the transmit power, power split ratio and subcarrier allocation to enhance the rate of IoT networks. In [21], a penalty function algorithm is proposed for energy-carrying communication networks to jointly optimize transmit power and collect energy to reduce system energy consumption. The authors of [22] investigate the problem of maximizing the throughput of IoT devices and propose a Lagrangian-based algorithm for this gradient, which jointly allocates spectrum and transmit power to improve the total system throughput. The authors of [23] address the energy efficiency optimization problem of optimized networks and propose an iterative algorithm based on the Karush–Kuhn–Tucker (KKT) condition to combine the transmit power and power coefficient, and its algorithm improves the energy efficiency performance of the system. In [24], a multi-objective energy-carrying network optimization problem was investigated, and the multi-objective problem was converted into a single-objective problem by using the defined equivalent sum-rate method for solving the problem, and the scheme optimized both system throughput and system energy consumption. In [25], a Dinkelbach-based two-layer iterative algorithm was proposed to jointly optimize the time-switching and power allocation problems. A particle swarm algorithm was proposed in [26] to optimize the rate of the SWIPT-enabled network.

Driven by the carbon-peaking and carbon-neutrality strategic goals, communication systems will move towards a “greener” direction, where balancing the requirements of low power consumption and high speed is crucial. The authors of [7,8,9,10] focus on the optimization of resource allocation with energy harvesting, which addresses the optimization of base station energy consumption and does not consider terminal standby time. The authors of [14] and others demonstrate IoT networks with energy cooperation, low power consumption and good performance. In [20,21,22,23,24,25,26] studied SWIPT networks optimized the on-net operation of terminals without introducing energy cooperation to save the power consumption of the H-AP. The above literature shows that SWIPT and energy cooperation technologies can effectively reduce system power consumption, however, the energy efficiency performance of SWIP-enabled IoT with energy cooperation is still unknown, and as people attach importance to the green network, energy efficiency becomes more and more important. Therefore, this paper aims to maximize the energy efficiency of the system and realize the design of a green communication system. Inspired by the aforementioned literature, we consider IoT networks in using energy cooperation and SWIPT to optimize the energy efficiency of the system as a goal. The main contributions of this paper are summarized as follows:We consider a downlink transmission model for SWIPT-enabled IoT with energy cooperation. A resource allocation problem is proposed that considers the quality of service (QoS) constraints for users, energy harvesting constraints, and jointly optimizes the power allocation, time switching coefficients and energy cooperation problems to maximize system energy efficiency as the optimization objective. Considering that the optimization problem is a mixed-integer non-linear programming problem that is difficult to solve directly, we consider decomposing the problem into three sub-problems of lower complexity, namely the power allocation, time switching coefficient and energy cooperation problems. We propose a two-stage algorithm for solving the problemThe first-stage algorithm is used to solve the power allocation and time-switching problems. The first-stage algorithm is a two-level iterative algorithm that the power allocation coefficient and the time switching coefficient are separated using the fixed variable method. In the outer layer, the power allocation solution is obtained using the Dinkelbach method iteratively. In the inner layer of the algorithm, the Dinkelbach method is used again to solve for the time switching coefficient under a fixed power allocation. The power allocation and time switching resolution are obtained by several iterations. Finally, in the second stage of the algorithm, the matching theory is used to obtain the resolution of energy cooperation.Our results show that our proposed algorithm has higher energy efficiency compared to the comparison algorithm. The system with SWIPT has good energy efficiency performance and can effectively extend the terminal on-grid operation cycle. In addition, the simulations show that energy cooperation can effectively reduce the energy consumption of the system. This demonstrates the performance benefits of integrating SWIPT and energy cooperation technologies in the IoT network.

The remainder of the paper is structured as follows. Section 2 presents the system-based model and the energy efficiency maximization problem modeling. In Section 3, we propose a two-layer algorithm to find the optimal joint power allocation and time-switching allocation. Section 4 investigates the use of a matching algorithm to solve the energy cooperation problem. Numerical results are given in Section 5 to prove the theoretical results. Finally, we conclude the paper in Section 6.

## 2. System Model

### 2.1. Transmission Model

As shown in Figure 1, we consider a SWIPT-enabled IoT system with energy cooperation, consisting of hybrid access points (H-AP) and a smart grid with downlink transmission, where the H-AP is fed by a mix of smart grid and renewable energy sources. The system consists of m H-AP and j terminal. Let m∈{1,2,3,⋯,M} denote the set of H-APs. Let j∈{1,2,3,⋯,N} denote the set of terminals. Each terminal contains information decoding and RF energy harvesting circuits. Considering the low cost and easy implementation of the time-switching method circuit, we distinguish between information and energy signals through the time-switching method. It is assumed that the H-AP is capable of serving multiple terminals simultaneously, the terminals associate with the nearest H-AP, the H-AP improves the spectrum efficiency of the system by sharing the entire transmission band, and multiple terminals under a single H-AP service use orthogonal spectrum resources, only one H-AP can be associated with a terminal. It is also assumed that all H-APs and terminals have perfect channel state information (CSI). The signal $S_{m}$ transmitted by the *m*-th H-AP can be expressed as $S_{m}=\sqrt{P_{jm}}s_{m}$, with $E\left[|s_{m}{|}^{H}\right]=1$, $P_{jm}$ indicates the transmitted power of the *m*-th H-AP. When the terminal is associated with the *m*-th H-AP, the received signal at the terminal can be expressed as
(1)$$y_{m}=h_{jm}\sqrt{P_{jm}}s_{m}+\sum_{m^{\prime }=1,m^{\prime }\neq m}^{M}h_{jm}^{m^{\prime }}\left(S_{m^{\prime }}\sqrt{P_{j^{\prime }m^{\prime }}}s_{m^{\prime }}\right)+\varpi_{0}$$
where, $\varpi_{0}$ denotes additive Gaussian white noise, and $h_{jm}\sqrt{P_{jm}}s_{m}$ denotes the accepted useful signal. $\sum_{m^{\prime }=1,m^{\prime }\neq m}^{M+1}h_{jm}^{m^{\prime }}\left(S_{m^{\prime }}\sqrt{P_{j^{\prime }m^{\prime }}}s_{m^{\prime }}\right)$ indicates received co-channel interference signal. The channel gain consists of path loss and Rayleigh fading, denoted as ${\left|h_{jm}\right|}^{2}=d^{-\beta }g$, g denotes Rayleigh fading, $d^{-\beta }$ denotes the path loss model, β denotes the path loss factor and *d* denotes the distance from the terminal to the H-AP. $P_{jm}$ indicates the transmit power from the H-AP *m* to the terminal j. $h_{jm}^{}$ denotes the channel gain of the terminal j associated with the H-AP m, $P_{jm}^{m^{\prime }}$ denotes the transmit power received by the terminal j from other H-APs and $h_{jm}^{m^{\prime }}$ denotes the channel gain of other H-AP interference.

$\gamma_{jm}$ is the signal interference noise ratio (SINR) of the terminals. Since the H-AP spectrum is shared within the network, there is mutual interference between the terminal links, and the SINR of the terminals is expressed as
(2)$$\gamma_{jm}=\frac{P_{jm}{\left|h_{jm}\right|}^{2}}{\sum_{m^{\prime }=1,m^{\prime }\neq m}^{M}\sum_{j^{\prime }=1}^{N}{\left|h_{jm}^{m^{\prime }}\right|}^{2}P_{j^{\prime }m^{\prime }}+\sigma^{2}}$$
where $\sigma^{2}$ is the noise power. $P_{jm}{\left|h_{jm}\right|}^{2}$ is the strength of the useful signal received by terminal j associated with H-AP m. $\sum_{m^{\prime }=1,m^{\prime }\neq m}^{M}\sum_{j^{\prime }=1}^{N}{\left|h_{jm}^{m^{\prime }}\right|}^{2}P_{j^{\prime }m^{\prime }}$ indicates that terminal j is subject to co-channel interference from terminals under other H-APs. Let ${\left|h_{jm}\right|}^{2}=G_{jm}$, ${\left|h_{jm}^{m^{\prime }}\right|}^{2}=G_{jm}^{m^{\prime }}$. The signal interference noise ratio (SINR) is expressed as
(3)$$\gamma_{jm}=\frac{P_{jm}G_{jm}}{\sum_{m^{\prime }=1,m^{\prime }\neq m}^{M}\sum_{j^{\prime }=1}^{N}G_{jm}^{m^{\prime }}P_{j^{\prime }m^{\prime }}+\sigma^{2}}$$

According to Shannon’s formula, the transmission rate of terminal j is given by
(4)$$R_{j}=\tau_{j}W\log_{2}(1+\gamma_{jm})$$
where *W* is the bandwidth of the system. $\tau_{j}$ is expressed as the transmission time allocated to the information time slot.

### 2.2. Energy Model

Assuming that all terminals collect energy from the RF signal using time-switching techniques and perform SWIPT techniques, the terminals split the received signal from the H-AP into two parts: in the first time slot for information transfer and in the second time slot for energy harvesting. Where $\tau_{j}$ is expressed as the transmission time allocated to the information time slot and $1-\tau_{j}$ is expressed as the time portion of the energy harvesting time slot. For the RF harvesting model of the IoT terminal, the paper uses the widely used linear energy harvesting model [25]. The energy harvesting circuit of the terminal is capable of converting the received power signal and the interfering power signal into DC power for use by the terminal. For the presence of multiple H-APs in an IoT system, the RF energy collected by the terminal consists of the superposition of the power emitted by multiple H-APs. The SWIPT-enabled terminals are used for energy harvesting within time slot $1-\tau_{j}$. The energy collected by a single terminal is expressed as
(5)$$E_{j}^{S}=(1-\tau_{j})\eta_{j}\sum_{m=1}^{M}G_{jm}P_{jm}$$
where $0<\eta_{j}<1$, $\eta_{j}$ denotes the conversion efficiency of the energy harvesting. It is assumed that the conversion efficiency of all the terminals of the system energy harvesting is the same, satisfying $\eta_{j}=\eta (\forall j)$.

In IoT systems with energy harvesting, each H-AP is equipped with an energy harvesting device. Due to the uneven distribution of renewable energy density and differences in transmitting power, some H-APs do not harvest enough energy to maintain their own standby power. Some H-APs harvest too much energy. To avoid wasting renewable energy, we have introduced energy cooperation technology, which is an important solution for effective energy dispatch. Through aggregators in the smart grid, the excess energy is transferred to the more power-consuming H-APs, effectively increasing the utilization of renewable energy. During the energy cooperation, the renewable energy received by H-AP is $\sum_{m=1}^{M}T_{m^{\prime }m}$, $\sum_{m=1}^{M}T_{mm^{\prime }}$ denotes the renewable energy transferred out of the H-AP. Where α∈[0,1] denotes the efficiency factor of the H-AP energy transfer. For a single H-AP, the available renewable energy is expressed as
(6)$$E_{m}=E_{m}^{RE}+\alpha \sum_{m=1}^{M}T_{m^{\prime }m}-\sum_{m=1}^{M}T_{mm^{\prime }}$$
where $T_{mm^{\prime }}$ represents energy transferred to other H-APs, $T_{m^{\prime }m}$ represents energy received from other H-APs and $E_{m}^{RE}$ is the renewable energy collected by the H-AP.

Typically, the power consumption of a conventional wireless communication system is defined as the following linear model [25], and the power consumption of an H-AP in a communication system is defined as the following model
(7)$$P_{W}=\zeta P_{m}+P_{m}^{C}$$
where ζ is the power amplification factor of the signal. $P_{m}^{C}$ is the standby power Consumption of the H-AP, including power consumption such as baseband signal processing and cooling system.

In IoT network, the circuit power consumption of SWIPT-enabled terminals is not considered because of their low device power consumption. The total energy consumption of the system is expressed as
(8)$$\sum_{m=1}^{M}G_{total}=\sum_{m=1}^{M}P_{W}-\sum_{m=1}^{M}E_{m}^{RE}-\sum_{j=1}^{N}E_{j}^{s}$$

### 2.3. Problem Formulation

The constraints in this paper include QoS constraints for terminals, maximum transmit power constraints for H-APs, and maximizing the total energy efficiency of the IoT system under energy harvesting constraints. According to [8], the energy efficiency (EE) of a system is defined as the ratio of the total achievable rate to the total power consumption. The optimization problem can then be expressed as
(9)$$\begin{array}{ll} P1:\;\;\mathrm{Max}:\; & \underset{P,\tau ,T}{EE(P,\tau ,T)}=\sum_{j=1}^{N}R_{j}/\sum_{m=1}^{M}G_{total}\\ s.t.\; & C1:\;R_{j}\geq R_{\min}\\ & C2:\;\sum_{j=1}^{N}E_{j}^{s}\geq E_{\min}\\ & C3:\;\sum_{j=1}^{N}P_{jm}\leq P_{m}^{\max}\\ & C4:\;T_{m^{\prime }m}\cap T_{mm^{\prime }}=\emptyset \\ & C5:\;0\leq P_{m}\;\;0\leq T_{mm^{\prime }}\\ & C6:\;0\leq \tau_{j}\leq 1 \end{array}$$
where $P=[P_{jm}]$, $\tau =[\tau_{j}]$, $T=[T_{mm^{\prime }}]$. C1 is the terminal minimum QoS requirement constraint. C2 represents the total terminal energy collection threshold under a single H-AP. C3, C5 denotes the transmit power constraint. C4 denotes that transfer of energy and reception cannot occur simultaneously. c6 denotes the constraint for the time switching factor.

We can observe that the **P1** problem is a complex fractional form, and the energy cooperation problem is an integer programming problem, then the P1 problem is a mixed-integer non-linear optimization problem that is difficult to solve directly. We consider reducing the problem to three less complex sub-problems, namely power allocation, time switching and energy cooperation. In addition the two sets of variables for the power allocation P and time switching coefficients τ, which are mutually coupled, as in the scheme in the literature [12], can be solved for the multi-variable problem by considering the remaining variables as constants and how to solve the remaining variables, and finally by an iterative scheme of alternating variables to obtain the resolution of the multi-variables. Because the power allocation P and time switching coefficients τ are coupled, we propose to solve the problem in the first stage using an iterative algorithm. In contrast, the subproblem of energy cooperation is uncoupled from the two sets of variables mentioned above, and for this reason we consider proposing a matching algorithm in the second stage to find the resolution of energy cooperation.

## 3. Joint Power Distribution and Time-Switching Control Algorithms

This section may be divided by subheadings. It should provide a concise and precise description of the experimental results, their interpretation, as well as the experimental conclusions that can be drawn.

### 3.1. Power Distribution Problem

In this section, we propose control algorithms for joint power allocation and time switching. From the P1 solution objective formulation, it is clear that optimizing the transmit power and time switching problems is the focus of ensuring terminal QoS and terminal standby time and improving system energy efficiency. Under the subproblem of ground-given energy collaboration. We have adopted the resource allocation scheme in [25], which is applied to the case of a single H-AP. We have expanded the scheme and applied it to multiple H-APs. we develop a two-layer iterative algorithm to solve for the power allocation and time-switching coefficients. Firstly, in the outer iteration, the iterative algorithm is used to solve for the transmit power given a time factor. In the inner layer of the algorithm, the transmit power is fixed and then the time switching coefficient is solved. The specific algorithmic analysis of the solution process is shown below.


*Power allocation method under timed switching allocation*


Given the two sets of variables for the time switching factor τ and energy cooperation T, only one set of variables for the transmit power needs to be solved. The original problem P1 is downscaled to solve a one-dimensional power distribution problem, then the **P1** problem is reformulated as
(10)$$\begin{array}{l} P2:\;\;\mathrm{Max}:\;EE(P)=\sum_{j=1}^{N}R_{j}/\sum_{m=1}^{M}G_{total}\\ \;s.t.\;C1,\;C2,\;C3,\;C5 \end{array}$$

As can be seen from the **P2** problem, the problem is a fractional objective function, making the problem non-linear and difficult to solve directly. First, we need to determine the non-concave nature of the optimization problem and then adopt the appropriate solution. The first step is to determine the non-concave nature of the constraints. Condition C1 can be converted to $\sum_{m^{\prime }=1,m^{\prime }\neq m}^{M}\sum_{j^{\prime }=1}^{N}G_{jm}^{m^{\prime }}P_{j^{\prime }m^{\prime }}+\sigma^{2}+P_{jm}G_{jm}-2^{\frac{R_{\min}}{\tau W}}(\sum_{m^{\prime }=1,m^{\prime }\neq m}^{M}\sum_{j^{\prime }=1}^{N}G_{jm}^{m^{\prime }}P_{j^{\prime }m^{\prime }}+\sigma^{2})\geq 0$. As can be seen from the inequality of the transformation deformation, the constraint is that the feasible domain of the resolution is a convex set. Similarly, the analytic feasible region of the constraint C2, C3, C5 is also a convex set.

Since $\underset{}{EE(P)}=\sum_{j=1}^{N}R_{j}/\sum_{m=1}^{M}G_{total}$ is fractional, we first prove the concavity of the numerator $\sum_{n=1}^{N}R_{j}$ of the objective function. Then, the first order derivative of $\sum_{n=1}^{N}R_{j}$ with respect to $P_{jm}$ is expressed as follows:(11)$$\frac{\partial \sum_{n=1}^{N}R_{j}}{\partial P_{jm}}=\frac{W\sum_{j=1}^{N}\tau_{j}}{\ln2}\cdot \frac{G_{jm}(\sum_{m^{\prime }=1,m^{\prime }\neq m}^{M}\sum_{j^{\prime }=1}^{N}G_{jm}^{m^{\prime }}P_{j^{\prime }m^{\prime }}+\sigma^{2})-P_{jm}G_{jm}\sum_{m^{\prime }=1,m^{\prime }\neq m}^{M}\sum_{j^{\prime }=1}^{N}G_{jm}^{m^{\prime }}}{\left(\sum_{m^{\prime }=1,m^{\prime }\neq m}^{M}\sum_{j^{\prime }=1}^{N}G_{jm}^{m^{\prime }}P_{j^{\prime }m^{\prime }}+\sigma^{2}+P_{jm}G_{jm})(\sum_{m^{\prime }=1,m^{\prime }\neq m}^{M}\sum_{j^{\prime }=1}^{N}G_{jm}^{m^{\prime }}P_{j^{\prime }m^{\prime }}+\sigma^{2}\right)}$$

Then, the second order derivative with respect to $P_{jm}$ is expressed as follows:(12)$$\begin{array}{l} \frac{\partial^{2}\sum_{n=1}^{N}R_{j}}{\partial P_{jm}P_{lm}}=-\frac{W\sum_{j=1}^{N}\tau_{j}}{\ln2}\cdot \frac{\left[G_{jm}(\sum_{m^{\prime }=1,m^{\prime }\neq m}^{M}\sum_{j^{\prime }=1}^{N}G_{jm}^{m^{\prime }}P_{j^{\prime }m^{\prime }}+\sigma^{2})-P_{jm}G_{jm}\cdot \sum_{m^{\prime }=1,m^{\prime }\neq m}^{M}\sum_{j^{\prime }=1}^{N}G_{jm}^{m^{\prime }}\right]}{{\left[\left(\sum_{m^{\prime }=1,m^{\prime }\neq m}^{M}\sum_{j^{\prime }=1}^{N}G_{jm}^{m^{\prime }}P_{j^{\prime }m^{\prime }}+\sigma^{2}+P_{jm}G_{jm})(\sum_{m^{\prime }=1,m^{\prime }\neq m}^{M}\sum_{j^{\prime }=1}^{N}G_{jm}^{m^{\prime }}P_{j^{\prime }m^{\prime }}+\sigma^{2}\right)\right]}^{2}}\\ \times \frac{\left[\left(\sum_{m^{\prime }=1,m^{\prime }\neq m}^{M}\sum_{j^{\prime }=1}^{N}G_{jm}^{m^{\prime }}+G_{jm})(\sum_{m^{\prime }=1,m^{\prime }\neq m}^{M}\sum_{j^{\prime }=1}^{N}G_{jm}^{m^{\prime }}P_{j^{\prime }m^{\prime }}+\sigma^{2})+\sum_{m^{\prime }=1,m^{\prime }\neq m}^{M}\sum_{j^{\prime }=1}^{N}G_{jm}^{m^{\prime }}\cdot (\sum_{m^{\prime }=1,m^{\prime }\neq m}^{M}\sum_{j^{\prime }=1}^{N}G_{jm}^{m^{\prime }}P_{j^{\prime }m^{\prime }}+\sigma^{2}+P_{jm}G_{jm}\right)\right]}{{\left[\left(\sum_{m^{\prime }=1,m^{\prime }\neq m}^{M}\sum_{j^{\prime }=1}^{N}G_{jm}^{m^{\prime }}P_{j^{\prime }m^{\prime }}+\sigma^{2}+P_{jm}G_{jm})(\sum_{m^{\prime }=1,m^{\prime }\neq m}^{M}\sum_{j^{\prime }=1}^{N}G_{jm}^{m^{\prime }}P_{j^{\prime }m^{\prime }}+\sigma^{2}\right)\right]}^{2}}\\ \forall j,l=1,2,\cdots N \end{array}$$

Let $H_{j}=\frac{\partial^{2}\sum_{n=1}^{N}R_{j}}{\partial P_{jm}{}^{2}}$, according to the above formula, we get
(13)$$\frac{\partial^{2}\sum_{n=1}^{N}R_{j}}{\partial P_{jm}P_{lm}}=\left\{ \begin{array}{l} H_{j},j\leq l\\ H_{l},\;\mathrm{otherwise} \end{array}\right.$$

Then, the Hessian matrix with respect to variables expressed as
(14)$$H=\left[\begin{array}{cccc} H_{1} & H_{1} & \cdots & H_{1}\\ H_{1} & H_{2} & \cdots & H_{2}\\ \vdots & \vdots & \ddots & \vdots \\ H_{1} & H_{2} & \cdots & H_{N} \end{array}\right]$$

Then, the opposite matrix of the Hessian matrix is Q=−H, then the *j*-th order principal subformula of the matrix is expressed as
(15)$$\begin{array}{l} Q_{j}=\left[\begin{array}{cccc} -H_{1} & -H_{1} & \cdots & -H_{1}\\ -H_{1} & -H_{2} & \cdots & -H_{2}\\ \vdots & \vdots & \ddots & \vdots \\ -H_{1} & -H_{2} & \cdots & -H_{N} \end{array}\right]\\ =\left\{ \begin{array}{l} -H_{1},j=1\\ -H_{1}\prod_{j=2}^{N}(H_{j-1}-H_{j}),2\leq j\leq N \end{array}\right. \end{array}$$

According to the formula, we can see that the power and channel is constantly greater than 0, and from the matrix properties can be obtained $H_{j-1}-H_{j}\geq 0$. Any *j*-order sequential principal subformula of matrix Q, $Q_{j}\geq 0$. It is possible to obtain Q≥0,H≤0. It follows that the Hessian matrix *H* of $\sum_{n=1}^{N}R_{j}$, with respect to the variable $P_{jm}$, is a semi-negative definite matrix. It follows that $\sum_{n=1}^{N}R_{j}$ is a concave function with respect to the variable $P_{jm}$. It follows from Shannon’s theorem that the communication rate is positive, thus it is shown that $\sum_{n=1}^{N}R_{j}$ is a non-negative concave function with respect to the variable $P_{jm}$. Similarly, the denominator of the objective function is non-negative. In summary, it is shown that the optimization objective function $\underset{}{EE(P)}=\sum_{n=1}^{N}R_{j}/\sum_{m=1}^{M}G_{total}$ is a concave fractional programming problem with respect to the transmit power P.

It can be seen that the objective function is a fractional programming problem, which is difficult to solve directly, and the Dinkelbach algorithm [26] has been widely used with solving non-linear fractional optimization problems. According to the nature of the Dinkelbach algorithm [27], the objective function needs to be transformed into the form of subtracting the numerator from the denominator. According to the Dinkelbach method, we need to introduce a parameter $\underset{}{e=EE(P)}$, and the P2 problem is converted into the following form
(16)$$\begin{array}{l} P2.1:\;\;\mathrm{Max}:\;\underset{}{EE(P)}=\sum_{n=1}^{N}R_{j}-e\cdot \sum_{m=1}^{M}G_{total}\\ \;s.t.\;C1,\;C2,\;C3,\;C5 \end{array}$$

**Proposition** **1.***Assuming that the optimal transmit power of the H-AP is*$P^{*}$*, the*$e^{*}$*is an optimal resolution of problem P2.1 for which the sufficient conditions are Max:*$F(e)=\sum_{n=1}^{N}R_{j}-e\cdot \sum_{m=1}^{M}G_{total}=0$.

**Proof.** The proposition is a classical conclusion in generalized dispersion planning, the proof of which has been proved in [28], and the proof process is not described in this paper. The above proposition provides the theoretical support for the transformation of the optimization problem **P2.1**. Thus, an approximate equivalent problem for problem P2.1 can be obtained by iteration. the Dinkelbach algorithm requires several iterations to obtain the resolution of the problem, where the power allocation and energy efficiency resolution at the *t*-th iteration are P(t) and e(t), respectively. Needs to be satisfied in the *t*-th iteration $F(e^{\left(t\right)})=\sum_{j=1}^{N}R_{j}(P^{\left(t\right)})-e^{\left(t\right)}\cdot \sum_{m=1}^{M}G_{total}(P^{\left(t\right)})≅0$. The Dinkelbach parameter $\underset{}{e}$ is updated by iterations until the convergence condition is met and the iteration is exited. From the above proof it can be seen that $\underset{}{EE(P)}=\sum_{j=1}^{N}R_{j}-e\cdot \sum_{m=1}^{M}G_{total}$ is a convex optimization problem and its solution function and constraints satisfy the scope of application of the Lagrangian dual method, so we can use the Lagrangian dual method to solve its optimization problem
(17)$$\begin{array}{l} L(P,\mu ,\nu ,\psi )=\sum_{j=1}^{N}\tau_{j}W\log_{2}\left(1+\frac{P_{jm}G_{jm}}{\sum_{m^{\prime }=1,m^{\prime }\neq m}^{M}\sum_{j^{\prime }=1}^{N}G_{jm}^{m^{\prime }}P_{j^{\prime }m^{\prime }}+\sigma^{2}}\right)\\ -e\left(\sum_{m=1}^{M}(\zeta P_{m}+P_{m}^{C}-E_{m})-\sum_{j=1}^{N}(1-\tau_{j})\eta_{j}\sum_{m=1}^{M}G_{jm}P_{m}\right)\\ -\mu_{j}\left(R_{\min}-\tau_{j}W\log_{2}\left(1+\frac{P_{jm}G_{jm}}{\sum_{m^{\prime }=1,m^{\prime }\neq m}^{M}\sum_{j^{\prime }=1}^{N}G_{jm}^{m^{\prime }}P_{j^{\prime }m^{\prime }}+\sigma^{2}}\right)\right)\\ -\nu_{j}\left(E_{\min}-\sum_{j=1}^{N}(1-\tau_{j})\eta_{j}\sum_{m=1}^{M}G_{jm}P_{m}\right)\\ -\psi \left(\sum_{j=1}^{N}P_{jm}-P_{m}^{\max}\right),\mu \geq 0,\nu \geq 0,\psi \geq 0 \end{array}$$
where denote the Lagrange multipliers of the constraints, respectively, and the pairwise function expressions are
(18)g(μ,ν,ψ)=maxL(P,μ,ν,ψ)The pairwise optimization problem expression for the problem is
(19)$$\begin{array}{l} \min g(\mu ,\nu ,\psi )\\ s.t.\;\mu \geq 0,\nu \geq 0,\psi \geq 0 \end{array}$$In this section, we use a gradient descent-based algorithm to obtain a power allocation solution to the pairwise optimization problem by multiple iterations, and we need to find the first-order derivative of the Lagrangian function [29], whose derivative is
(20)$$\begin{array}{l} \frac{\partial L}{\partial P_{jm}}=\frac{W\sum_{j=1}^{N}\tau_{j}}{\ln2}\cdot \frac{G_{jm}(\sum_{m^{\prime }=1,m^{\prime }\neq m}^{M}\sum_{j^{\prime }=1}^{N}G_{jm}^{m^{\prime }}P_{j^{\prime }m^{\prime }}+\sigma^{2})-P_{jm}G_{jm}\sum_{m^{\prime }=1,m^{\prime }\neq m}^{M}\sum_{j^{\prime }=1}^{N}G_{jm}^{m^{\prime }}}{\left(\sum_{m^{\prime }=1,m^{\prime }\neq m}^{M}\sum_{j^{\prime }=1}^{N}G_{jm}^{m^{\prime }}P_{j^{\prime }m^{\prime }}+\sigma^{2}+P_{jm}G_{jm})(\sum_{m^{\prime }=1,m^{\prime }\neq m}^{M}\sum_{j^{\prime }=1}^{N}G_{jm}^{m^{\prime }}P_{j^{\prime }m^{\prime }}+\sigma^{2}\right)}\\ +\frac{\mu_{j}W\tau_{j}}{\ln2}\cdot \frac{G_{jm}(\sum_{m^{\prime }=1,m^{\prime }\neq m}^{M}\sum_{j^{\prime }=1}^{N}G_{jm}^{m^{\prime }}P_{j^{\prime }m^{\prime }}+\sigma^{2})-P_{jm}G_{jm}\sum_{m^{\prime }=1,m^{\prime }\neq m}^{M}\sum_{j^{\prime }=1}^{N}G_{jm}^{m^{\prime }}}{\left(\sum_{m^{\prime }=1,m^{\prime }\neq m}^{M}\sum_{j^{\prime }=1}^{N}G_{jm}^{m^{\prime }}P_{j^{\prime }m^{\prime }}+\sigma^{2}+P_{jm}G_{jm})(\sum_{m^{\prime }=1,m^{\prime }\neq m}^{M}\sum_{j^{\prime }=1}^{N}G_{jm}^{m^{\prime }}P_{j^{\prime }m^{\prime }}+\sigma^{2}\right)}\\ -e\left(\zeta -\sum_{j=1}^{N}(1-\tau_{j})\eta_{j}\sum_{m=1}^{M}G_{jm}P_{m}\right)+\nu_{j}\left((1-\tau_{j})\eta_{j}\sum_{m=1}^{M}G_{jm}\right)-\psi \end{array}$$Based on the Lagrangian derivative of the dual function, we give an updated formula for the power distribution, expressed as follows:(21)$$P_{jm}(t+1)={\left(P_{jm}(t)+\beta \frac{\partial L}{\partial P_{jm}}(t)\right)}^{+}$$To ensure that the iteration values converge, we update the step size to satisfy $\beta (t)=\frac{\beta (t-1)}{t-1}$. We use the subgradient method to update the Lagrange multipliers of the constraints [25,30]. The subgradient formulation of the dual function g(μ,ν,ψ) is given by
(22)$$\begin{array}{l} \frac{\partial g}{\partial \mu_{j}}=\tau_{j}W\log_{2}\left(1+\frac{P_{jm}G_{jm}}{\sum_{m^{\prime }=1,m^{\prime }\neq m}^{M}\sum_{j^{\prime }=1}^{N}G_{jm}^{m^{\prime }}P_{j^{\prime }m^{\prime }}+\sigma^{2}}\right)-R_{\min}\\ \frac{\partial g}{\partial \nu_{j}}=\sum_{j=1}^{N}(1-\tau_{j})\eta_{j}\sum_{m=1}^{M}G_{jm}P_{m}-E_{\min}\\ \frac{\partial g}{\partial \psi_{j}}=P_{m}^{\max}-\sum_{j=1}^{N}P_{jm} \end{array}$$The Lagrange multiplier update equation is as follows:(23)$$\begin{array}{l} \mu_{j}(t+1)={\left(\mu_{j}(t)+\delta \cdot \frac{\delta g}{\delta \mu_{j}}\right)}^{+}\\ \nu_{j}(t+1)={\left(\nu_{j}(t)+\delta \cdot \frac{\delta g}{\delta \nu_{j}}\right)}^{+}\\ \psi (t+1)={\left(\psi (t)+\delta \cdot \frac{\delta g}{\delta \psi }\right)}^{+} \end{array}$$
where δ is the iteration step. □

### 3.2. Time Switching Problem

This section analyzes the time-switching scheme in detail. In the inner layer of the algorithm, the transmit power is fixed and then the time switching coefficients are solved by iteration. After fixing the transmit power P, the time switching coefficients τ are a set of unknown variables to be solved, then the problem is reformulated as
(24)$$\begin{array}{l} P2.2:\;\;\mathrm{Max}:\;\underset{}{EE(\tau )}=\sum_{j=1}^{N}R_{j}/\sum_{m=1}^{M}G_{total}\\ \;s.t.\;C1,\;C2,\;C6 \end{array}$$

To simplify the expression of the formula we make
(25)$$\begin{array}{l} A=W\log_{2}\left(1+\frac{P_{jm}G_{jm}}{\sum_{m^{\prime }=1,m^{\prime }\neq m}^{M}\sum_{j^{\prime }=1}^{N}G_{jm}^{m^{\prime }}P_{j^{\prime }m^{\prime }}+\sigma^{2}}\right)\\ B=\sum_{m=1}^{M}(\zeta P_{m}+P_{m}^{C}-E_{m})-\eta_{j}\sum_{j=1}^{N}\sum_{m=1}^{M}G_{jm}P_{m}\\ C=\eta_{j}\sum_{j=1}^{N}\sum_{m=1}^{M}G_{jm}P_{m} \end{array}$$

It can be seen that the problem is a fractional programming problem with time coefficients, and as above, we also use the Dinkelbach method for the fractional problem, for which the problem is reformulated after the Dinkelbach treatment as
(26)$$\begin{array}{c} P2.3\;\;\;\;\mathrm{Max}:\;\underset{}{F(\tau )}=\sum_{j=1}^{N}A\tau_{j}-\lambda (B+C\sum_{j=1}^{N}\tau_{j})\\ \;s.t.\;A\tau_{j}\geq R_{\min}\\ C(1-\tau_{j})\geq E_{\min} \end{array}$$

It is clear that the problem is a one-time programming problem with a one-time switching coefficient τ, and it is only necessary to prove whether the function is monotonic to find the resolution of the time switching factor. The derivative of the problem **P2.3** is
(27)$$\frac{\partial F(\tau )}{\partial \tau }=A-\lambda C$$

If the derivative is positive, then the function is monotonically increasing, then the time switching coefficient τ for its positive direction of the boundary value, if the negative, the function is monotonically decreasing, then the minimum value of the boundary value for the negative direction. The time switching coefficient discriminant is expressed as follows:(28)$$\tau^{*}\left\{ \begin{array}{l} \max\{0,1-\frac{E_{\min}}{C}\},if\;A-\lambda C\geq 0\\ \min\{1,\frac{R_{\min}}{A}\},if\;A-\lambda C<0 \end{array}\right.$$

We also consider a special scenario. When the terminal is close to the H-AP, the throughput is almost similar. In this case, we can think that the time switching coefficients are the same, and the computational complexity can be reduced. Finally, we obtain the resolution of the power distribution and the time switching coefficients by iteration of the alternating variables method. The two-level iterative algorithm for power allocation and time switching is shown in Algorithm 1.
**Algorithm 1.** Two-layer iteration for power allocation and time switching1: **Input:** transmit power P, time switching coefficient τ, Lagrange multiplier μ,ν,ψ, Maximum number of iterations $T_{out}$, Update step β(t),δ(t), QoS threshold $R_{\min}$, threshold $E_{\min}$ number of iterations *t* = 1, and ε;2: **Output:** energy efficiency EE(P,τ)
3: **for** j=1:1:N
4: **for** 1<t<Tout
5:  update transmit power Pjm(t+1) according to (21)6:  update Lagrange multiplier μ,ν,ψ according to (23)7:  **for** 1<t<Tout8:   update time switching coefficient τ according to (28)9:  **end for**10:    **if** EE(t+1)-EE(t+1)≤ε exit loop;11:     **break**12:    **end if**13: **end for**14: **end for**

## 4. Energy Cooperation Programmer

In this section, we investigate IoT systems with energy cooperation scenarios. We develop a many-to-many matching algorithm to solve the renewable energy dispatch problem. As can be seen from the system diagram, energy cooperation is done through aggregators in the smart grid, which act as intermediaries between the base stations and the grid, so that the grid operator charges a fee for the energy exchange through the aggregators, but it is lower than the actual fee, due to the fact that the renewable energy is acquired by the base stations [31]. The issue of cost is beyond the scope of this paper. In this paper, we only consider the scheduling of incoming energy from the perspective of energy consumption and quality of service optimization.

When given two sets of variables using the transmit power P and time switching coefficients τ, only one set of variables for the energy cooperation T needs to be solved. The original optimization problem P1 can be reformulated as optimization problem P4, which is formulated as follows:(29)$$\begin{array}{c} P3\;\max EE(T)\\ \;\;s.t.\;C4,\;C5 \end{array}$$

The energy cooperation subproblem is an integer combinatorial optimization problem. In this paper, we consider the use of many-to-many matching theory to solve it. Matching theory is one of the effective solution tools for studying decentralized resource allocation and can transform the resource allocation problem into a simple distributed problem. In existing studies, the many-to-many matching theory has been used to solve optimization problems related to wireless networks [15,32,33], demonstrating that matching theory has the characteristics of fast convergence and stable configuration results.

We adopt the allocation scheme based on the matching theory in [15,32]. We extend the scheme to the IoT network and adjust the preference according to the characteristics of the IoT network. According to the matching theory allocation scheme, the H-APs are first divided into two sets of categories, where the set $M^{+}=\{m^{+}\in M|E_{m}^{RE}-P_{W}>0\}$ denotes the set of H-APs that have excess energy while satisfying their own power consumption, both in terms of energy output. $M^{-}=\{m^{-}\in M|E_{m}^{RE}-P_{W}<0\}$ indicates a collection of H-APs that do not collect enough energy to sustain their own power consumption. Based on the principle of matching bilateral benefits [15], the utility functions (preference degrees) corresponding to each other between the two types of H-APs are established, and setting a suitable preference degree function can effectively reduce the loss of collected energy and improve the energy utilization of the system. The matching diagram of its two types of collections. Each H-AP in set $M^{+}$ corresponds to all H-APs in set $M^{-}$ and has a preference list corresponding to it. Similarly, the base stations in set $M^{-}$ have their own preference for the H-APs in set $M^{+}$. The premise of matching is to first match the base station with the corresponding base station according to its preference to complete the energy cooperation.

In the process of energy cooperation, the transmission efficiency is mainly related to the resistance value of the power line, the greater the resistance value, the greater the energy loss, the loss of energy is expressed as follows:(30)$$\;E_{loss}=I^{2}R(l)$$
where I is the current in the transmission line, R(l) is the total resistance of the power line, R(l)=ρl, ρ is the resistance factor and l is the length of the power line [34]. It can be seen that the lost energy is positively related to the length of the power line. The transmission efficiency $\alpha_{mm^{\prime }}$ from H-AP $m^{+}$ to H-AP $m^{-}$ is expressed as
(31)$$\;\alpha_{mm^{\prime }}=\frac{T_{mm^{\prime }}-E_{loss}}{T_{mm^{\prime }}}$$
where $T_{mm^{\prime }}$ indicates the renewable energy allocated by H-AP $m^{+}$ to H-AP $m^{-}$. There is a difference in transmission efficiency $\alpha_{mm^{\prime }}$ between the two H-APs due to the different lengths of the power lines. Set $M^{+}$ prefers to transfer energy to a base station with higher transmission efficiency to reduce energy losses, so the preference of H-APs in Set $M^{+}$ over those in Set $M^{-}$ is expressed as
(32)$$p(M^{+},M^{-})=\alpha_{mm^{\prime }},\;m^{-}\in M^{-}$$

When H-AP $m^{-}$ sends an energy request to the H-APs in Set $M^{+}$, the H-APs in set $M^{+}$ will select the H-APs with the highest ranking according to the preference ranking in (32) and accept its request, passing the energy to H-APs $m^{-}$.

H-Aps $M^{-}$ within the set prefer H-APs with more energy remaining in set $M^{+}$, as this reduces the number of passes and responses from the H-AP and the H-AP is able to obtain energy faster. The preference of the H-APs in set $M^{-}$ for the H-APs in set $M^{+}$ is expressed as
(33)$$p(M^{-},M^{+})=E_{m}^{RE}-\zeta P_{m}-P_{m}^{C},\;m^{+}\in M^{+}$$

H-AP $m^{-}$ is ranked according to the preference of (33) and H-AP $m^{-}$ selects the H-AP with the highest preference in set $m^{+}$.

Based on the previous analysis, we propose a solution of joint power allocation, time switching and energy cooperation. The specific solution process is summarized in Algorithm 2.
**Algorithm 2.** Joint power allocation, time switching and energy cooperation algorithm$0:\;\mathrm{Input}:\;\mathrm{transmit}\;\mathrm{power}\;P,\;\mathrm{time}\;\mathrm{switching}\;\mathrm{coefficient}\;\tau ,\mathrm{Lagrange}\;\mathrm{multiplier}\;\mu ,\nu ,\psi ,\;\mathrm{maximum}\;\mathrm{number}\;\mathrm{of}\;\mathrm{iterations}\;T_{out},\;\mathrm{update}\;\mathrm{step}\;\beta (t),\;\delta (t),\;\mathrm{QoS}\;\mathrm{threshold}\;R_{\min},\;\mathrm{threshold}\;E_{\min}\;\mathrm{number}\;\mathrm{of}\;\mathrm{iterations}\;t=1,\;\mathrm{and}\;\varepsilon$;1: Output: energy efficiency EE(P,τ,T)2: According to Algorithm 1, obtain transmit power P, time switching τ solution3: for m=1:1:M$4:\;\mathrm{Calculate}\;\mathrm{the}\;\mathrm{preference}\;\mathrm{of}\;\mathrm{all}\;\mathrm{the}\;H-\mathrm{AP}\;\mathrm{in}\;\mathrm{set}\;M^{+}$ according to (32), and rank them$5:\;\mathrm{Calculate}\;\mathrm{the}\;\mathrm{preference}\;\mathrm{of}\;\mathrm{all}\;\mathrm{the}\;H-\mathrm{AP}\;\mathrm{in}\;\mathrm{set}\;M^{-}$ according to (33), and rank them $6:\;H-\mathrm{AP}\;m^{+}$$\mathrm{selects}\;\mathrm{the}\;H-\mathrm{AP}\;\mathrm{with}\;\mathrm{the}\;\mathrm{largest}\;\mathrm{preference}\;\mathrm{in}\;\mathrm{set}\;M^{-}$ to complete the energy cooperation.$7:\;\;\mathrm{if}\;\mathrm{the}\;\mathrm{set}\;M^{-}$$\mathrm{or}\;M^{+}$ is the empty set, exit loop;8:    **break**9:  **end if**10: **end for**

### Complexity and Convergence Analysis

The joint algorithm consists of a two-stage algorithm. Based on the computational complexity of the Dinkelbach method $O(\frac{1}{\varepsilon^{2}}\log(K))$ [25]. Both the inner and outer iteration processes use the Dinkelbach method. Thus, the computational complexity of the proposed two-layer iterative algorithm is approximately $O(\frac{1}{\varepsilon_{1}^{2}}\frac{1}{\varepsilon_{2}^{2}}\log(K))$. The complexity of the matching algorithm is related to the length of the two sets, with a complexity of $O(M_{1}M_{2})$, the complexity of the joint algorithm is $O(\frac{1}{\varepsilon_{1}^{2}}\frac{1}{\varepsilon_{2}^{2}}\log(K)+(M_{1}M_{2}))$.

## 5. Performance Analysis

This section verifies the effectiveness of the algorithm through simulation. It is assumed that there are 5 H-APs in the network. The cell range of the H-APs is 100 × 100 m^2^, the terminals *N* = 10 is uniformly distributed within a 10m radius of the H-AP with a terminal RF energy conversion rate of 0.5 [25]. The communication rate threshold is 10 Mbit/s, terminal energy collection thresholds is 0.01 mW. The channel fading model contains Rayleigh fading and path loss, the channel gain is denoted as $d^{-\beta }g$, where $d^{-\beta }$ is the path loss, β = 2, and g is the small-scale fading, generated by the Rayleigh distribution, with a mean difference of 0 and a variance of 1 [19]. The static power consumption is 6 W and the energy transfer efficiency is rand (0.7–0.9) [12]. The joint optimization problem solved by the proposed algorithm following algorithms to compare the performance of each aspect: the rate-maximization (Max-rate) algorithm, which optimizes the power allocation and time switching [25]. The wireless portable energy resource optimization algorithm, which optimizes power allocation and time switching using particle swarm optimization (PSO) algorithms [26]. The energy cooperation part adopts the matching algorithm in this paper. The simulation parameters are shown in Table 1.

Figure 2 shows the iterative convergence of the algorithm. It can be seen from the figure that the algorithm proposed in this paper is the fastest in terms of convergence speed and also obtains the highest energy efficiency performance. As can be seen from the algorithm flow diagram, the algorithm in this paper is given certain initial values in the setting of parameters, such as energy efficiency and firing power, such that the initial values will reduce the number of iterations to some extent. The particle swarm algorithm, on the other hand, searches from a global resolution, so the search resolution requires a large number of iterations to complete, and therefore iterative convergence becomes slower. This suggests that, for this system, our proposed algorithm, has some performance advantage in terms of convergence speed.

Figure 3 shows the curve of the number of terminals versus energy efficiency. As can be seen from the figure, our proposed algorithm achieves higher energy efficiency compared to the PSO algorithm and the maximum rate algorithm. This is because our algorithm achieves a higher resolution of power allocation during the convergence iterations, which effectively suppresses the co-channel interference to its users, which results in a higher throughput for the users, and at the same time, the system consumes less transmit power. The particle swarm algorithm, on the other hand, tends to fall into localized resolution and does not have high search accuracy. The maximum rate algorithm, although able to obtain higher rates, consumes more transmit power, resulting in a less energy efficient system. On the other hand, it can be seen from the figure that the use of energy cooperation techniques can improve the energy efficiency of the system compared to scenarios where no energy cooperation techniques are used, because the excess renewable energy is fully utilized and the consumption of the grid is reduced. As can be seen from the figure, the algorithm of this paper can be effectively applied to a multi-terminal scenario.

Figure 4 shows the curve of the effect of the number of H-APs on energy consumption. From the figure, it can be seen that as the number of H-APs increases, the energy consumption of the system also increases, which is due to the increase in static power consumption of the H-APs. The algorithm in this paper and the PSO algorithm both use energy cooperation technology, which makes full use of renewable energy to reduce the system energy consumption. H-APs that do not use energy cooperation will consume more energy. The maximum rate algorithm has the highest energy consumption because, in order to obtain a higher throughput, the transmit power is high, which leads to more energy consumption of the system on the grid. The algorithm proposed in this paper consumes less energy than the PSO algorithm because the algorithm achieves a resolution that is closer to the optimal solution through multiple iterations and consumes less transmit power. As a result, the algorithm is more suitable for multi-H-AP scenarios.

Figure 5 shows the relationship between the number of H-APs and energy efficiency. As can be seen from the figure, the energy efficiency decreases as the number of H-APs increases. This is due to the fact that the standby energy consumption of the system increases as the number of H-APs increases, resulting in a decrease in energy efficiency. On the other hand, the algorithm proposed in this paper outperforms other algorithms in terms of energy efficiency. This is because our proposed algorithm achieves higher throughput and lower energy consumption in the power allocation problem, and therefore higher energy efficiency performance, which indicates that our proposed algorithm can be applied to multi-H-APs scenarios.

Figure 6 shows the effect of QoS on the energy efficiency of the system. The graph shows that as the QoS increases, the energy efficiency of the system decreases. This is because, in order to ensure the QoS constraint, the H-APs needs to increase the transmitting power to ensure the QoS demanded by terminals with poor channel conditions, which increases the energy consumption of the system and consequently decreases the energy efficiency. The graph shows that the energy efficiency of the system decreases more slowly when the QoS is increased compared to other algorithms, which indicates that this paper has a higher resolution accuracy in solving the power allocation and therefore achieves a higher energy efficiency than other algorithms.

Figure 7 shows the curve of the effect of energy harvesting on the energy efficiency of the system. As can be seen from the figure, the energy efficiency of the system decreases as the energy harvesting constraint increases. This is because, in order to satisfy the energy harvesting constraint, on the one hand, the H-APs needs to transmit more power to satisfy the constraint, which increases the energy consumption of the system and leads to a decrease in energy efficiency. On the other hand, in order to satisfy the energy collection constraint, the time for the energy collection part will increase, while the time for the message decoding will decrease accordingly, which will increase the amount of energy collected by the terminal, but at the same time will decrease the throughput of the terminal. The introduction of the SWIPT technology, which increases the standby time of the terminal, the effect of this technology on the energy efficiency of the system is negligible, is very promising. There is a trade-off between throughput and terminal standby time, and in practical scenarios the values can be set according to the different terminal categories.

Figure 8 gives the curve of the effect of the number of H-APs on the energy consumption of the system. As can be seen from the graph, as the number of H-APs increases, the energy consumption of the system also increases. This is because the energy consumption increases as the standby power consumption of the H-APs increases, and although energy harvesting techniques are introduced, they are not yet able to balance the standby energy consumption. In this section, we mainly show the performance comparison between our proposed energy cooperation algorithm and the comparison algorithm DES [15]. It can be seen from the figure that our algorithm performs better due to the impact of the transmission efficiency that we prioritize in setting the matching preference, which, to a certain extent, reduces the energy loss during transmission. On the other hand, the graph shows that the two sets of algorithms that introduce energy cooperation significantly outperform the scenario without energy cooperation in terms of energy consumption. The energy cooperation technique makes full use of renewable energy sources and avoids the waste of excess energy. Therefore, the introduction of energy cooperation techniques has a positive effect on the energy consumption of the communication system.

Figure 9 shows the effect of the number of terminals on the energy collected. It is clear from the figure that as the number of terminals increases, the amount of energy collected also increases. It is clear that based on the PSO algorithm its obtained energy collection is significantly better than the algorithm proposed in this paper, this is because the particle swarm algorithm has a larger value of power allocation, although the impact on the energy consumption and energy efficiency of the H-APs is negative, this increases the amount of energy collected, which is positive for the standby time of the terminals. The present algorithm, on the other hand, mainly optimizes the energy efficiency of the system and therefore obtains a smaller resolution of the power allocation, which is able to suppress co-channel interference and increase the throughput, but the amount of energy collected by the terminal is then reduced. Additionally, for different systems under the system, it is necessary to weigh the H-AP power consumption and terminal standby time and set the energy collection constraint of the terminal for different needs.

## 6. Conclusions

In this paper, we study the SWIPT-enabled IoT network with energy cooperation. We develop a mathematical model with energy efficiency as the optimization objective, while needing to satisfy quality of service and minimum energy harvesting constraints. The problem is nonlinear and difficult to solve directly. We propose an iterative algorithm to solve the problem of power allocation, time switching and energy cooperation. Simulation results show that. Our proposed algorithm outperforms the comparison algorithm in terms of energy efficiency performance. Moreover, this algorithm has good performance for multi H-APs and multi terminal scenarios. In addition, simulation shows that SWIPT technology can effectively extend the operation cycle of the terminal, and the energy cooperation technology can effectively reduce the system energy consumption, which is positive for the development of green communication.

Our algorithm can be extended to other networks with energy collection, especially low-power terminal device networks, such as the current research focus on 5G networks with NOMA, or heterogeneous network systems, where the energy efficiency of the system can be effectively improved using our proposed algorithm for systems with multiple H-APs. In future work, there is still room for improvement. In this paper, we consider the case of having perfect channel conditions; according to the literature [34], the case of imperfect channel conditions information can cause interruptions and rate degradation, for such problems still need further analysis and proposed solutions. On the other hand, with the rapid development of smart grids and energy cooperation technology involving the trading of energy in order to weigh the interests of energy intermediaries and communication operators, further research is needed to solve it.

## Figures and Tables

**Figure 1 sensors-22-05035-f001:**
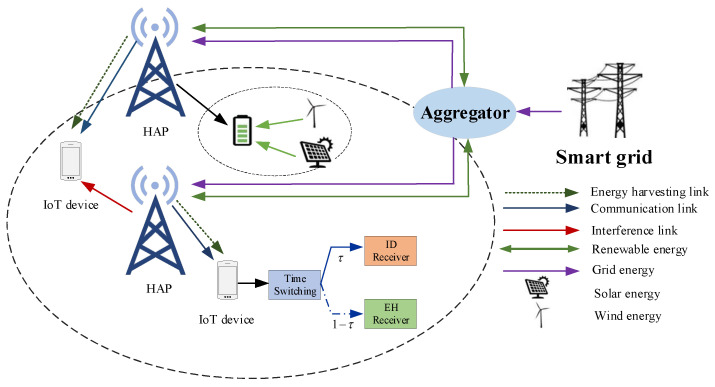
System model.

**Figure 2 sensors-22-05035-f002:**
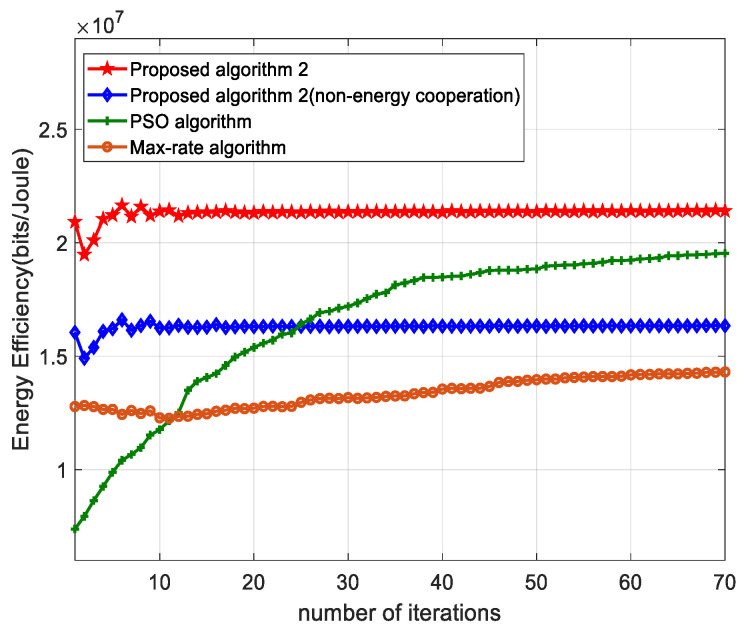
Convergence performance of different algorithm.

**Figure 3 sensors-22-05035-f003:**
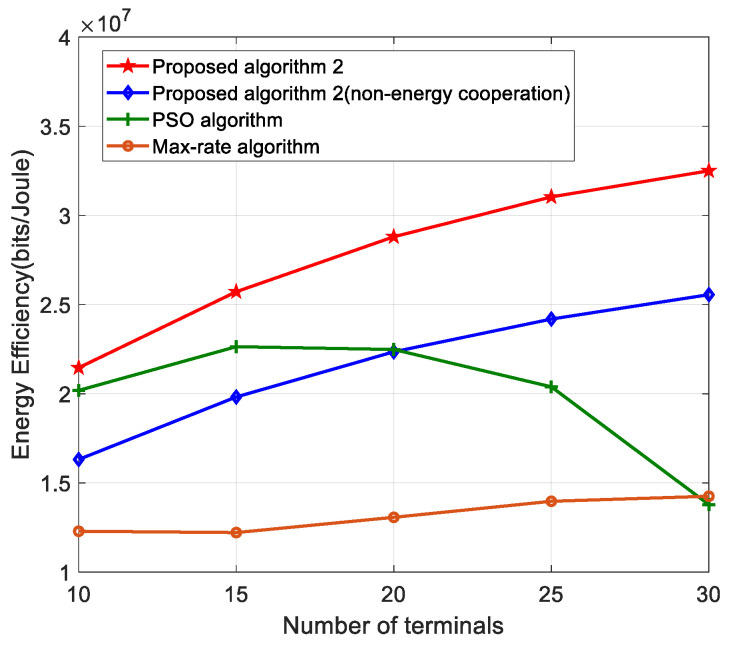
Energy efficiency versus number of terminals for different algorithms.

**Figure 4 sensors-22-05035-f004:**
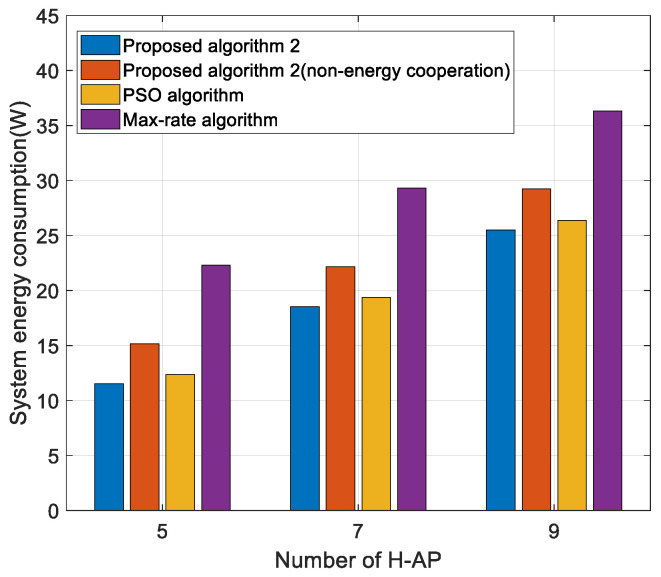
System energy consumption versus the number of H-APs for different algorithms.

**Figure 5 sensors-22-05035-f005:**
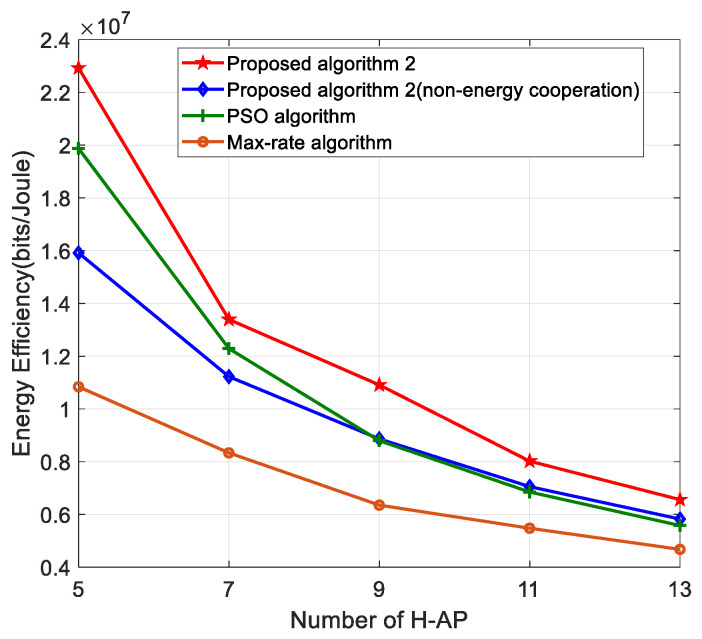
Energy efficiency versus the number of H-APs for different algorithms.

**Figure 6 sensors-22-05035-f006:**
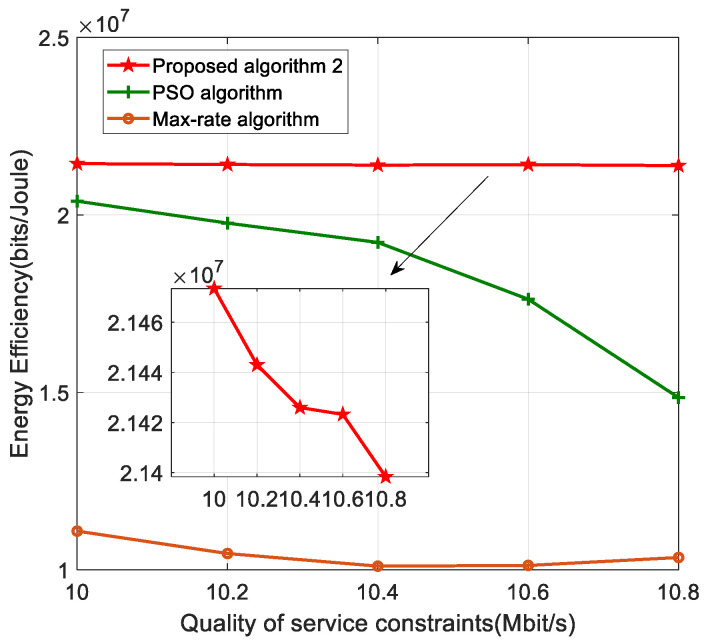
Energy efficiency versus the QoS for different algorithms.

**Figure 7 sensors-22-05035-f007:**
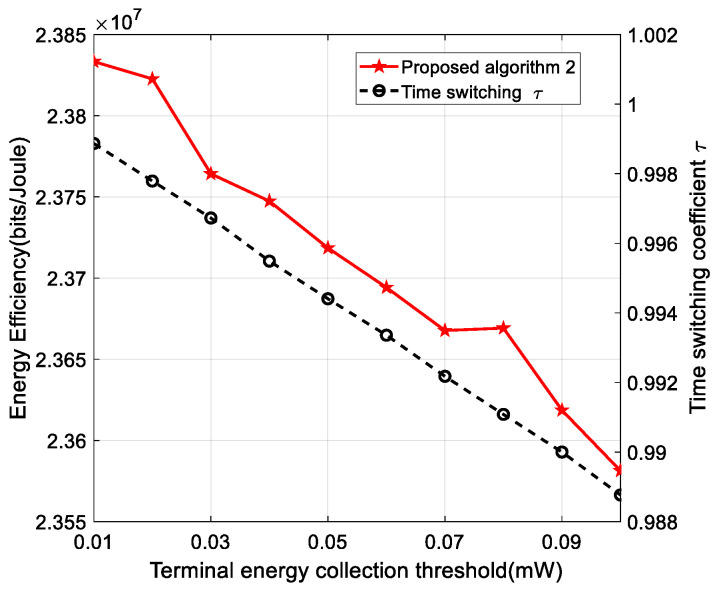
The performance of the proposed algorithm with different collection threshold.

**Figure 8 sensors-22-05035-f008:**
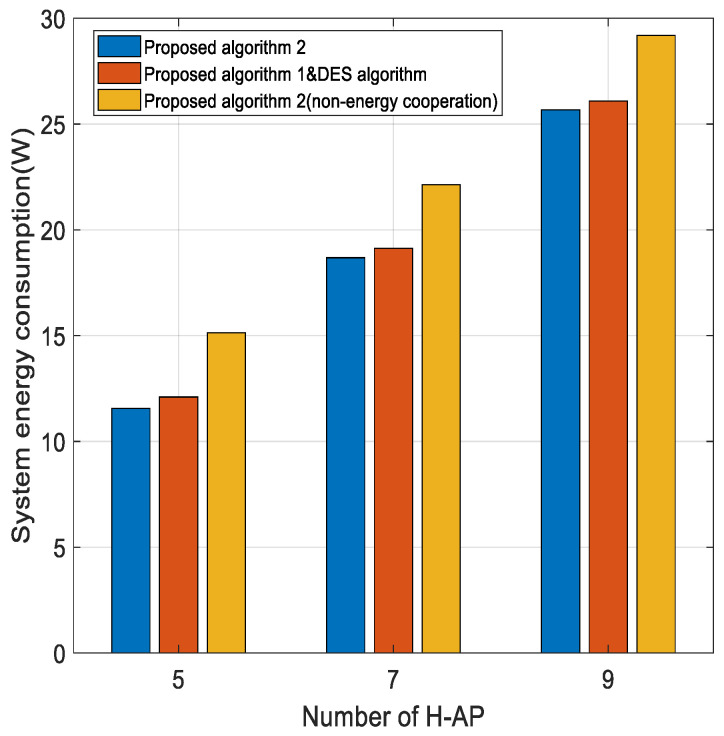
System energy consumption versus the number of H-AP for different algorithms.

**Figure 9 sensors-22-05035-f009:**
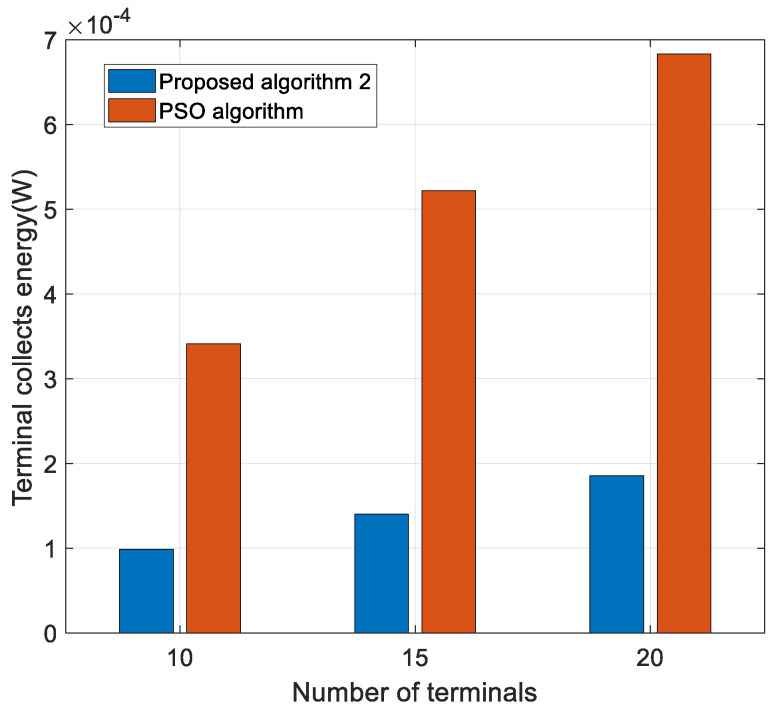
Terminal collects energy versus the number of terminals for different algorithms.

**Table 1 sensors-22-05035-t001:** System Parameters.

Parameter	Value
System bandwidth	10 MHz
Noise power density	−174 dBm/Hz
Max transmit power of H-AP	30 dBm
transmit power generation factor	$\zeta =\frac{1}{38\%}$
H-AP Energy collection	4–10 W

## Data Availability

Not applicable.

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
