# Peer review of "Energy Efficiency Optimization for SWIPT-Enabled IoT Network with Energy Cooperation"

_sensors, 2022, doi:10.3390/s22135035_

Round 1

Reviewer 1 Report

First, congratulations on writing such a valuable article. I would also like to thank you for your dedication to writing the article.

However, I propose to improve the article, in particular, the analysis of the literature taking into account the items from the MDPI publishing house.

1. The structure of the paper is good. First, the introduction presents the theme briefly. Then, the paper has a literature review section and a problem description section.

2.   The objective and contribution could be written more clearly in the introduction section.

3.        The literature analysis presented in the paper can be improved. It was prepared briefly and does not take into account all the aspects relevant to the scope of research included in the article.

4.        propose to include articles from the MDPI publishing house in the literature analysis.

5.        Lines 246 and 247

The correct notation is: C2, C3, C4, C5.

6.        Conclusions should be expanded and supplemented with research results.

7.         The proposed amendments will increase the substantive value of the article.

Reviewer 2 Report

In this paper, the authors studied the and SWIPT-enabled IoT networks with energy cooperation, in which a mathematical model with energy efficiency as the optimisation objective is formulated. The topic is interesting. However, the following issues need to be reconsidered.

1. How to gunratee the HAP perform energy cooperation?

2. What does \tau_j and P_m mean in (4) and (5), respectively?

3.  The authors need to prove why the constraints satisfy the scope of application of the Lagrangian dual method.

4. Is it realistic to assume \beta equals to 2?

5. In [25], the algorithm is performed with NOMA scheme, is it fair to compare the performance with it?

Reviewer 3 Report

The paper contributes to the pressing field of energy efficiency in SWIPT networks. The work presented is technically sound and the authors have presented significant mathematical proofs supporting their contributions. 

In terms of my comments, I would suggest authors rigorously proof read the paper to avoid typos and inconsistencies in terms of acronyms, citation procedure and grammar. There is also some repetition in the text which needs to be avoided to improve quality. 

One of my major concern is regarding the defined energy model. Authors claim that the circuit power consumption of SWIPT-enabled terminals is not considered because of their low device power consumption. However, in IoT networks most of the end-device circuits contain sensors that require significant power. I would suggest adding a reference to justify this decision. 

Round 2

Reviewer 1 Report

The authors addressed all my comment properly.

Reviewer 2 Report

No more comments.